# Controversies in Diagnosing Sarcopenia in Cirrhosis—Moving from Research to Clinical Practice

**DOI:** 10.3390/nu11102454

**Published:** 2019-10-14

**Authors:** Marie Sinclair

**Affiliations:** 1Department of Medicine, The University of Melbourne, Parkville 3050, Australia; marie.sinclair@austin.org.au; Tel.: +6-139-496-5353; Fax: +6-139-496-3487; 2Austin Health, Liver Transplant Unit, 145 Studley Road, Heidelberg 3084, Australia

**Keywords:** sarcopenia, cirrhosis, diagnosis, muscle, mortality, transplant

## Abstract

Sarcopenia, defined as loss of muscle mass and function, is increasingly recognized as a common consequence of advanced cirrhosis that is associated with adverse clinical outcomes. Despite the recent proliferation in publications pertaining to sarcopenia in end-stage liver disease, there remains no single ‘best method’ for its diagnosis. The inability to identify a gold standard is common to other specialties, including geriatrics from which many diagnostic tools are derived. Controversies in diagnosis have implications for the accuracy and reproducibility of cohort studies in the field, largely prohibit the introduction of sarcopenia measurement into routine patient care and impede the development of clinical trials to identify appropriate therapies. Difficulties in diagnosis are partly driven by our ongoing limited understanding of the pathophysiology of sarcopenia in cirrhosis, the mechanisms by which it impacts on patient outcomes, the heterogeneity of patient populations, and the accuracy, availability and cost of assessments of muscle mass and function. This review discusses the currently studied diagnostic methods for sarcopenia in cirrhosis, and outlines why reaching a consensus on sarcopenia diagnosis is important and suggests potential ways to improve diagnostic criteria to allow us to translate sarcopenia research into improvements in clinical care.

## 1. Introduction

According to the European Working Group on Sarcopenia in Older People, sarcopenia is defined as the loss of each of muscle mass, muscle strength and reduced physical function [1]. Sarcopenia can therefore exist independently of cachexia, or muscle wasting. Sarcopenia in specific chronic disease populations, including liver disease, has often been appropriated in the literature as a reduction in muscle mass alone for the purpose of population-based prognostic studies [2,3]. This however does not provide the full clinical picture, as muscle mass does not always correlate well with muscle strength or function in either the geriatric or the cirrhotic population [4,5]. The end-result of sarcopenia is functional decline, estimated by physical frailty measures derived in geriatric populations and subsequently validated in cirrhosis [6]. Existing studies have interchangeably assessed these three parameters, being muscle mass, muscle strength, and physical frailty; creating difficulty in comparing results between studies.

The traditional condition of sarcopenia as described in the literature was as a slowly progressive age-related phenomenon that occurs due to replacement of muscle fibres with fat, increased muscle fibrosis, oxidative stress and metabolic alterations, in conjunction with deterioration of the neuromuscular junction. Sarcopenia of ageing is acknowledged to be a multifactorial process that occurs due to changes in physical activity, hormone levels, nutrition, mitochondrial dysfunction and changes in systemic inflammation [7]. Although some of these factors also exist in cirrhosis, they are known to be specific disease-related factors that contribute to muscle loss in the cirrhotic population [8].

The prevalence of sarcopenia approaches 70% in many reports of cirrhotic patients awaiting liver transplantation [8] but prevalence varies widely depending on the diagnostic technique employed, the specific population studied, and patient gender. For example, patients with NAFLD have a lower rate of sarcopenia than other disease aetiologies (22% vs. 47%, *p* < 0.001) but relatively higher rates of frailty (49% vs. 34%, *p* = 0.03) [9], which may relate to a relative as opposed to absolute deficiency in muscle mass in comparison to fat mass.

This article provides a brief overview of sarcopenia in cirrhosis and aims primarily to discuss the controversies in the diagnosis of sarcopenia, how this impacts the interpretation of the current literature and describes the limitation of current diagnostic tools in both research and clinical care, and suggests future directions for sarcopenia research.

## 2. What Causes Sarcopenia in Cirrhosis?

As in the geriatric population, the pathogenesis of sarcopenia in cirrhosis is multifactorial, however there are specific factors unique to liver disease. It is clear that the cirrhotic population is highly heterogenous, with differing contributing factors that likely relate to underlying differences in disease aetiology, severity of portal hypertension and gender-specific hormone imbalances [8]. The end-result is a progressive loss of muscle mass and function due to imbalances in muscle formation and muscle breakdown and alterations in energy requirements.

Reduced muscle formation in cirrhosis is contributed to by reduced oral intake due to nausea, reduced gastric reserve and dysgeusia as well as macronutrient malabsorption related to portal hypertension and biliary conditions [10]. Circulating serum branched chain amino acids are reduced in cirrhotic patients due to their uptake in muscle to clear elevated circulating ammonia via glutamine synthase, depriving muscle of its preferential fuel [11]. Serum testosterone in males and insulin-like growth factor-1 (IGF-1) across many cirrhotics are reduced, removing positive stimuli of new muscle formation, as well as contributing to upregulation of myostatin, which further inhibits myogenesis [8,12,13].

Increased muscle breakdown occurs in patients with cirrhosis due to reduced hepatic glycogen stores, resulting in increased lipid utilization and protein catabolism even during short periods of fasting such as overnight [14]. Mitochondrial dysfunction has been reported to contribute to muscle autophagy, which is potentially mediated by hyperammonemia [15], and there is increased activation of the ubiquitin protease pathway, which may relate to both inactivity and elevated systemic inflammation that are observed in cirrhotic patients [16]. Energy requirements are also elevated in some patients with advanced cirrhosis compared to healthy controls [17], potentially due to a combination of increased systemic inflammation and heat loss due to peripheral vasodilation and the presence of ascites, which further compounds muscle loss.

A full discussion as to potential contributing factors has been well-explored in previous literature [8,15] and is outside the scope of this article. The impact of different causes of sarcopenia on an individual remains unknown. Whether such differing factors have a different clinical course or prognosis remains unknown, and it may also be that differing aetiological factors may also lead to variations in response to proposed treatments of sarcopenia in an individual.

## 3. What Methods Are Currently Used to Diagnose Sarcopenia?

There is currently no gold standard for the diagnosis of sarcopenia in any population, including liver disease. Current methods tend to estimate one feature of sarcopenia, such as muscle mass or function in a specific muscle group. Few studies of muscle mass report on muscle quality and those that analyse functional assessments often utilize differing protocols in differing populations. Key techniques that have been studied in cirrhosis are summarized below with current evidence for their use in liver disease summarized in Table 1.

### 3.1. Nutritional Assessment

The main nutritional assessment tool employed and analyzed in cirrhosis is the Subjective Global Assessment, or SGA. This comprises a questionnaire usually administered by qualified dieticians and assesses nutritional intake, change in weight, functional capacity and physical examination. Patients are classified as either well nourished (A), mild-to-moderately malnourished (B) or severely malnourished (C) [43]. Moderate and severe malnutrition using the SGA is common in cirrhosis (exceeding 50%) [9] and increases in conjunction with severity of liver disease [18].

### 3.2. Anthropometry

Traditional anthropometric measures including weight and body mass index have fallen out of favor, given their inability to differentiate fat from muscle. Specifically, in liver cirrhosis, fluid accumulation can significantly impact on anthropometry, including BMI measures and lower limb measures. However, several key measures are still used in clinical care and have also been assessed in clinical studies of patients with cirrhosis. The predominant measures currently studied in cirrhosis focus on the upper limb to avoid confounding by fluid and include mid-arm muscle circumference (MAMC), as well as triceps skinfold thickness [21].

### 3.3. Functional Measures

Many functional measures have been proposed to assess sarcopenia, with handgrip strength the most widely used and well-studied across all populations. Handgrip strength is usually performed with a calibrated dynamometer using the non-dominant hand and averaged over three successful attempts, and has been widely studied in cirrhosis, with defined normal ranges in both men and women [44]. The short physical performance battery was initially derived in geriatric populations, and assesses balance, gait speed and chair stands with a score given out of 12, and has been studied in cohorts of patients awaiting liver transplant [26]. The Fried Frailty Index is an additional tool initially derived in the geriatric population that has since been modified in to a liver-specific tool called the Liver Frailty Index after analysis in a large cohort of cirrhotics awaiting liver transplant [27]. This index identified the combination of grip strength, chair stands and balance as being the best predictors of mortality in cirrhosis and from this data the researchers created an equation with a score ranging from 0 to 7, with <3.2 classed as robust and >4.5 classed as frail. The final commonly studied functional measure in cirrhosis is the six min walk test, which simply assesses the distance walked in 6 min at the patient’s usual walking pace [28].

### 3.4. Bioelectrical Impedance

Bioelectrical impedance (BIA) is a non-invasive technique that measures the passage of electricity through body tissues to estimate the body composition. The principal on which it is based is that water leads to lower impedance, and more muscular patients tends to have more body water than patients with high adiposity. The device can measure fat-free mass by estimating total body water, but is subjective to confounding by both states of dehydration, recent exercise and fluid retention [45]. There has been considerable interest in the use of BIA in subjects with cirrhosis to quantify muscle mass given it is safe, simple to measure and relatively cheap [46], however little work has been done to validate its accuracy in patients with cirrhosis.

### 3.5. Quantitative Measures of Muscle Mass

Ultrasound is the simplest, cheapest tool that can be employed to assess muscle mass, with most studies examining its use assessing the quadriceps muscle. Quadriceps muscle thickness and quality has been proposed as an objective measure of sarcopenia that has been validated in a cohort of ICU patients [47]. In a cohort of 159 subjects with cirrhosis, quadriceps thickness using ultrasound had reasonable concordance with cross-sectional imaging in its ability to identify patients with sarcopenia [48], with the obvious benefit of being radiation-free, accessible and cheap. Cross-sectional imaging is by far the most widely assessed measure of sarcopenia in subjects with cirrhosis, and the test of choice is the L3 skeletal muscle index, which is the muscle area on a CT scan at the level of the L3 vertebrae corrected for height [30]. This obviously requires a small amount of radiation to perform, but also requires additional computer software to accurately calculate the muscle area on a single slice CT scan. There is however, as yet, no protocol to standardize the CT scan at the L3 level in regards to the exact slice location and width, to minimize variability between scans. Additional CT based measures included psoas muscle diameter, which requires no specialised computer software to analyse, and psoas muscle area. There is however, less data on the use of these parameters in the cirrhotic population as compared to the L3 skeletal muscle index with conflicting reports as to their significance [3,35].

Dual energy X-ray absorptiometry (DEXA) body composition uses low-dose X-rays to provide a comprehensive 3-dimensional analysis of the entire body and automatically breaks down each body compartment into bone mass, fat mass and fat-free (or lean mass). The radiation exposure is minimal; however, cost and access are an issue in some parts of the world. Although the readings are highly reproducible with a coefficient of variation <0.5% [49], the lean mass reading cannot differentiate muscle from water, raising issues in patients with cirrhosis who have ascites and peripheral oedema. To minimize confounding by fluid, both appendicular (arms and legs) and upper limb lean mass have been proposed as tools to analyse muscle mass in cirrhosis [38,41]. Finally, magnetic resonance imaging (MRI) is considered to be an appealing test for the diagnosis of muscle wasting due to the lack of radiation exposure and high quality images, including information on muscle quality as evidence by fat infiltration [50]. At present however, both cost and access issues preclude the routine use of MRI for the diagnosis of sarcopenia for clinical purposes, and there remains little data on its use in liver disease.

## 4. Why Is Assessing Sarcopenia in Cirrhosis Important?

It is now well reported in the hepatology literature that sarcopenia is associated with poor outcomes in patients with cirrhosis. Documented adverse outcomes include mortality (HR 2.21, *p* = 0.008), sepsis-associated mortality (22% in sarcopenic versus 8% in non-sarcopenic patients, *p* = 0.02) and acute-on-chronic liver failure (HR ranging from 3.4 to 6.8, *p* < 0.01) [2,3,51]. Sarcopenia appears to be more common in men than women with cirrhosis and may portend greater clinical risk in men [2], however more functional measures estimating frailty have clearly been associated with mortality (50% increase in wait-list mortality, *p* = 0.01) as well as drop-out from the liver transplant waitlist (22% vs. 10%, *p* = 0.03) in both genders [6]. Previous studies may have been underpowered to assess sarcopenia risk in women, and a larger multi-centre series did indeed demonstrate mortality risk associated with sarcopenia in women, using a cut-off of 39 cm^2^/m^2^ for L3 skeletal muscle index [32]. Mortality risk from sarcopenia appears to extend to patients with hepatocellular carcinoma (HR 1.69–1.74, *p* = 0.006), and in one study of patients on sorafenib, remained significant independent of tumor burden and severity of liver disease (HR 1.61, *p* = 0.03) [52,53]. Frailty has additionally been associated with increased hospitalization days in patients awaiting liver transplantation (IRR 1.21, *p* = 0.03) [54].

The data on clinical outcomes post-liver transplantation are more ambiguous, with conflicting findings pertaining to mortality. A significant association has been observed using psoas muscle diameter to diagnose sarcopenia (HR 3.7, *p* < 0.001) [55] and in men only using single slice CT at the L3 level (HR 0.95, *p* = 0.011 for each unit increase in muscle mass for men), with no association observed for women [56]. Conversely, a relatively large study of 248 patients employing single slice CT at the L3 level found no relationship between sarcopenia and post-transplant survival (survival 117 ± 17 months vs. 146 ± 20 months, *p* = 0.4), however did show a significantly increased length of stay in sarcopenic patients (40 ± 4 days vs. 25 ± 3 days, *p* = 0.005) [57]. This is similar to a smaller study of 96 patients that showed no significant mortality risk but increased rates of postoperative complications in patients with sarcopenia (40.4% vs. 18.4%, *p* = 0.01) [58]; however, such a study is likely underpowered to assess mortality risk.

Post-transplant mortality risk clearly needs further clarification given that some studies have shown that incorporating sarcopenia into prognostic algorithms can reduce liver transplant waitlist mortality. The currently employed tool, the Model for End-stage Liver Disease (MELD) score, purely assesses liver disease severity, without taking into account other systemic features. A MELD-sarcopenia score, incorporating CT-measured sarcopenia, has been shown to better predict liver transplant waitlist mortality as compared to the MELD score alone, particularly in patients with a low MELD score (c-statistic 0.85 vs. 0.69, *p* = 0.02 in patients with MELD <15) [59]. It remains unknown, however, whether there is a critical muscle mass or level of physical function below which transplantation may be futile. It is also important to recognize that sarcopenia tends to worsen in patients on the liver transplant waitlist (average reduction in grip strength of 0.38 kg per 3 months) [25] and thus patients should be monitored serially for deterioration.

## 5. What Do We Want in a Diagnostic Test for Sarcopenia?

Most importantly, an ideal test must be accurate, in that it needs to be highly reproducible if multiple measurements are performed in an individual at a single time point. It also needs to be able to accurately compare difference between individuals and be accurately repeated in serial measurements over time. Without such precision, comparing change over time in an individual or between individuals is not possible and will not add useful information to improve clinical decision making. There is surprisingly little data on the accuracy of various methods of sarcopenia diagnosis in any population, let alone the cirrhotic literature.

It has been shown that different observers calculating muscle area from a single slice CT scan on a single patient with cirrhosis have a high intraclass coefficient correlation (>0.99), suggesting excellent reproducibility [60], however we do not know if repeating the CT scan in the same patient and then re-analysing has a similar accuracy, as such studies have not been performed. The plane of slice, the thickness and the exact level of the slice could each potentially impact on the reproducibility of this test and these factors have not been well studied, neither is there a universally accepted protocol for conducting these scans. Most studies examining the use of single slice CT to quantify sarcopenia in cirrhosis have only been used at a single point in time [2]. A retrospective post-TIPS (transjugular intrahepatic portosystemic shunt) series that demonstrated survival benefit with attenuation of sarcopenia used single slice CT to compare change in muscle over time [61], but there was no clear protocol in place to ensure exact reproducibility of the single slice, which could impact on muscle area measurement. A small retrospective study of serial CT scans in a population with adrenocortical carcinoma used computer-based algorithms to compare changes in psoas muscle cross-sectional area over time, and although such technology is likely to improve reproducibility this method has not been externally validated [62].

Serial scans on an individual have been performed in healthy young adults using DEXA body composition for the purpose of evaluating accuracy of measurement, and demonstrated a high precision, with a coefficient of variation of only 0.5% [39]. DEXA lean mass and frailty measures have been administered serially in cirrhotic studies [25,40], but the accuracy of these repeated measurements in the cirrhotic population are difficult to validate given the lack of gold standard. Functional measures require patient cooperation and have subjective components and may therefore be prone to inaccuracy and difficult to standardize. A recent study of the Liver Frailty Index however performed two separate tests of frailty on the same patient in a single day and suggests that handgrip strength has excellent reproducibility (intraclass correlation coefficient (ICC) 0.93 (0.90–0.94)), with lower reproducibility for chair stands (ICC 0.87 (0.83–0.90) and balance (ICC 0.73 (0.65–0.79)). Potential confounders include changes in fluid retention confounding DEXA lean mass, and musculoskeletal complaints, fatigue or encephalopathy impacting on functional measures.

Reproducibility for serial testing is a requirement for any test to be considered for a MELD-sarcopenia score or for the purpose of quantifying response to therapeutic interventions. Reproducibility requires a clear protocol for test administration and validation studies to confirm its use as a serial measurement. Regardless of the accuracy and reproducibility of a single test, it also needs to be accessible, affordable and safe. Many patients with cirrhosis live in regional areas [63], that may have little access to personnel trained at administering frailty measures, and technology to quantify muscle mass. CT scans and DEXA scans are also expensive and difficult to access in some regions. Although DEXA confers incredibly low radiation doses (0.001 mSv (millisieverts)), CT scans can administer up to 15 mSv of ionizing radiation [64]. Dose exposure has been reduced with modern scanning software, however the radiation conferred by CT scans has implications, if they are to be used for serial measurements, with cumulative exposure to ionizing radiation being associated with a relative lifetime cancer risk increase of up to 26.5% [65].

Finally, an ‘ideal’ sarcopenia test needs to have clinical value, in that it predicts clinically significant endpoints such as mortality, infection or hospitalization. Almost all forms of sarcopenia measurement have been linked to clinical outcomes [2,3,6,28,41,54], however few studies have compared the utility of different techniques. In the geriatric literature, there is a suggestion that functional measures may be superior to muscle mass in respect to predicting mortality [66]. In the liver literature, our group found handgrip strength to be a superior predictor of pre-transplant mortality as compared to CT and DEXA body composition in a retrospective series of 145 men waitlisted for liver transplant [24]. This finding is similar to a previous study that found functional but not CT-based measures of muscle mass were associated with waitlist mortality [5], but few other such studies have been performed. Clearly, a larger multi-centre study examining multiple diagnostic methods using clear diagnostic protocols is required to better determine which diagnostic test (or tests) holds the most clinical value in patients with cirrhosis. It also remains to be determined, whether different diagnostic techniques should be used in differing aetiologies such as NAFLD [9] and whether gender-specific tools are required.

## 6. Is Sarcopenia a Binary Phenomenon?

There are few guidelines that recommend specific diagnostic criteria for sarcopenia in cirrhosis. However, a large north American consensus statement has recently suggested sarcopenia is best defined in cirrhosis using the skeletal muscle index at the L3 level using CT scan. This statement defined sarcopenia as <50 cm^2^/m^2^ in men, and <39 cm^2^/m^2^ in women [30]. There is however, little data to support this recommendation and it remains unknown whether having a specific cut-off value for sarcopenia is even the best way to utilize muscle mass measurements.

The prevalence of sarcopenia dramatically changes according to the choice of test and cut-off used for diagnosis, with higher rates of sarcopenia seen when CT modalities are employed [20,24]. It has been claimed, that a higher rate of diagnosis of sarcopenia is preferable [20], but it there is no evidence that diagnosing a higher proportion of patients with sarcopenia improves outcome prediction, and in fact diagnosing sarcopenia by any modality is associated with mortality risk despite the marked differences in sarcopenia prevalence [24]. Cut-offs within a diagnostic modality have also differed between studies, with some cut-offs applied to cirrhotics extrapolated from other populations, for example CT L3 muscle cut-off of <52.4 cm^2^/m^2^ for men and <38.5 cm^2^/m^2^ for women used in initial cirrhotic work was defined in oncology populations [2]. More recent studies have performed cut-point analyses on datasets of cirrhotics and identified population-specific cut-offs, defined as <50 cm^2^/m^2^ for men and <39 cm^2^/m^2^ for women for CT L3 [32], <1.6 kg/m^2^ for upper limb lean mass by DEXA [41] and <19.5 kg for handgrip strength [67].

It is unclear whether examining sarcopenia as a binary phenomenon is the most appropriate way to analyse muscle mass or strength. Presence or absence of sarcopenia clearly impacts on mortality in cirrhosis [2,68], but it may be that there is a linear relationship between muscle mass and clinical outcomes. Studies analysing muscle mass or strength as continuous variables have identified progressive improvement with each unit change; each 1 kg increase in handgrip strength correlated with a 6% reduction in mortality [24], a 1 unit decrease in the Fried Frailty Score was associated with a 45% increase in mortality [6] and a 1 unit reduction in height-adjusted psoas muscle diameter was been associated with a 15% increase in mortality risk [3]. Given the heterogeneity of such studies regarding diagnostic modality and patient population, it is uncertain whether there a continuous relationship purely exists within a specific clinical range of muscle mass or function, whether there is a threshold above or below which this relationship no longer exists, and how severity of liver disease interacts with sarcopenia risk. Establishing this relationship is vital if a sarcopenia tool is ever to be used for a formal MELD-sarcopenia score.

## 7. Does Treating Sarcopenia Improve Outcome?

Given the known adverse associations of sarcopenia, the presumption is that improving muscle mass and function improves clinical outcomes, however this is not supported by evidence, largely due to the lack of large-scale randomized controlled trials of successful therapies for sarcopenia. There are no medications or interventions specifically approved to treat sarcopenia in cirrhosis and it is not yet known what specific dietary and exercise recommendations are most effective. The strongest evidence for mortality benefit for treating sarcopenia comes from a retrospective study of patients undergoing TIPS stent placement for portal hypertension, that demonstrated a significant reduction in mortality in the subset of patients for whom sarcopenia resolved (9.8% mortality as compared to 43.5% in patients with unchanged muscle mass, *p* = 0.007) [61]. However, this was not a randomized study and fails to address other potential factors that could have resulted in both muscle gain and reduction in mortality risk.

Although diet and exercise are frequently recommended to combat sarcopenia in cirrhosis, there are no studies that demonstrate a mortality benefit from any specific regimen [69]. There is evidence to suggest that dietary interventions may improve muscle mass, total body protein and quality of life [70,71,72], and exercise interventions have been shown to increase exercise capacity, VO2 max and fatigue scores [73,74], however none of these studies demonstrated a mortality benefit. This may be because many of these studies were inadequately powered to assess mortality as an outcome, and larger multi-centre trials are still required. A 2018 Cochrane review into exercise interventions for cirrhosis found no mortality benefit, however again the quality of evidence was deemed to be low [75].

A randomized placebo-controlled trial of testosterone therapy in 101 men conducted by our group demonstrated a significant improvement in muscle mass in treated men, with a trend to reduced mortality in the treatment arm (16% vs. 25.5%) however this did not meet significance (*p* = 0.352) and the study was not powered to determine this outcome [40]. This lack of evidence does not imply that treating sarcopenia is futile, but that few effective interventions are available, and larger scale, multi-centre RCTs are required. These studies need to assess interventions proven to improve muscle mass and function, large enough to take into account the multiple potential confounders in a heterogenous cirrhotic population, to better determine whether sarcopenia reversal does indeed improve clinical outcomes. 

## 8. Where to Next in Sarcopenia Research?

Prospective large-scale, multi-centre cohort studies are required comparing multiple diagnostic methods of sarcopenia using an accepted standardized methodology with complete collection of data clinical outcomes both pre and post-liver transplant. Ideal such a study should include novel techniques such as proteomics biomarkers analysis or computer-algorithm derived methods of imaging analysis. A large-scale analysis should allow for diagnostic criteria to be formed for both men and women and may also identify differences that need to be applied to differing disease aetiologies or ethnicities. Such studies should allow us to better predict prognosis in our patients with cirrhosis and identify those in need of intervention.

The establishment of accepted diagnostic tools will better allow us to measure outcomes of sarcopenia interventions, both clinically and for the purpose of randomized controlled trials. The few existing RCTs for sarcopenia and malnutrition in cirrhosis have used very different endpoints ranging from protein turnover to CT muscle mass and DEXA lean mass [14,40,61]. Large-scale multi-centre RCTs of promising therapies (such as testosterone, branched chain amino acids, TIPS or myostatin blockers) should ideally use an accepted diagnostic modality to measure response. Such studies should also be adequately powered to ascertain whether improving muscle or function translates to improved clinical outcome, including mortality, infection, hospital length of stay and quality of life.

From a transplant perspective further consideration needs to be given to a MELD-sarcopenia score [59,68], that has been shown to improve prediction of waitlist mortality as compared to MELD score alone (using CT L3 and diagnosis sarcopenia in a binary fashion). It is imperative that a sarcopenia tool for this purpose be frequently reproducible to allow for dynamic adjustment in conjunction with the MELD score, have strong prognostic value, and be reliably standardized across different centers to be applicable for nation-wide allocation programs. Finally, we need some guidance as to whether there is a degree of sarcopenia which renders patients “too sick for transplant” to avoid futile transplantation and allow organ allocation to those most likely to benefit. Such a cut-off would ideally be identified from large scale observational cohorts with well-characterized post-transplant outcomes.

## 9. Conclusions

Despite the clear advances in the recognition of the importance of sarcopenia there remains no clear gold standard for its diagnosis. Wide variations exist between the prevalence and impact of sarcopenia using different diagnostic methods and there are no standardized protocols for existing tests. This uncertainty limits the application of sarcopenia diagnosis into clinical care and makes clinical trial design difficult. Further research is clearly required to better delineate the best way to measure muscle mass and function to allow us to use this important information to improve patient care.

## Figures and Tables

**Table 1 nutrients-11-02454-t001:** Summary of existing diagnostic techniques for sarcopenia in cirrhosis.

Mode of Diagnosis	Defined Cut-Off for Sarcopenia in Cirrhosis	Correlation with Pre-Transplant Mortality	Pros	Cons	Suitability for Serial Measurement	Future Research Needs
Subjective global assessment (SGA) Class A, B or C	A = well nourished B = mild-moderately malnourished C = severely malnourished	Mortality was higher in SGA class C (47%) as compared to SGA A (3.1%) in a cohort of 100 cirrhotics [18] Mortality was higher in SGA C patients (HR 9.4 (0, 26.2), *p* = 0.01) in a cohort of 315 patients awaiting liver transplant, however this lost significance on multivariable analysis [19]	Cheap Non-invasive, safe No specialised equipment or software required	Subjective components Requires a trained dietician to administer Little differentiation between degrees of malnutrition	High	Validation in a broader cohort of patients with cirrhosis is required to determine independent prognostic value
Mid-arm muscle circumference (MAMC) (cm)	<10th percentile for age and gender [20]	Sarcopenia as defined by MAMC was associated with reduced survival at 6, 12 and 24 months in 212 hospitalized cirrhotics (*p* < 0.001) [21]	Safe Readily available Inexpensive Rapid Not affected by oedema	Not well validated Cannot differentiate fat mass from muscle mass or fluid retention	High	Needs further studies in broader cohorts of patients, and to ensure no confounding from fluid retention
Handgrip strength (HGS) (kg)	Men: <30 kg Women: <15 kg [22] OR Men: <26 kg Women: <18 kg [23]	HGS was associated with mortality in men (HR 0.96, *p* < 0.01) and women (HR 0.91, *p* = 0.02) in a cohort of 563 patients with cirrhosis [22] Each 1 kg increase in HGS reduced mortality by 6% in a cohort of 145 men assessed for liver transplant [24], and each 1 kg reduction in HGS in an individual while awaiting liver transplant increased mortality by 11% in a cohort of 309 transplant candidates [25]	Simple Rapid Inexpensive Reproducible	May be affected by patient effort, hepatic encephalopathy and musculoskeletal comorbidities	High	Consensus on measurement protocol requirement (e.g., dominant vs. non-dominant hand) Further validation investigation variation/natural fluctuation over time required
Short physical performance battery (SPPB) Score out of 12	Frail is defined as a score ≤9	Frailty is associated with increased waitlist mortality (HR 2.36, *p* = 0.01) only in patients >65 years of age [26]	Simple Safe Minimal training required to administer	Potential confounding by musculoskeletal complaints or other comorbidities	High	Needs further studies in broader cohorts of cirrhotics
Liver Frailty Index score (range 0 to 7) (based on gender, HGS, balance and chair stands)	Frail: ≥4.5 [27]	c-index for mortality for the MELDNa-Frailty index was 0.82 versus 0.80 for the MELDNa score alone in a cohort of 536 cirrhotics, suggesting improved prognostic value using the Frailty index [27]	Safe Inexpensive Readily available	May be affected by patient effort, hepatic encephalopathy and musculoskeletal comorbidities	High	Needs external validation in other cohorts
6-min walk test (6MWT) (metres)	<250 m [28]	Mortality was reduced by 52% for every 100 m increase in the baseline 6MWT in a study of 121 cirrhotics listed for transplant. Distance walked inversely associated with mortality (HR 0.48, *p* = 0.001) [28]	Simple, no specific training or equipment to administer Functional test Cheap	May be affected by patient effort, hepatic encephalopathy and musculoskeletal comorbidities	High	Larger cohort studies to validate in broader cohorts of cirrhotics
Bioelectrical impedance (BIA) Appendicular skeletal muscle index (ASMI—kg/m^2^) Or Phase angle (degrees)	ASMI Men: <7.0 kg/m^2^ Women: <5.7 kg/m^2^ [23] Phase angle <4.9 degrees	Phase angle <4.9 degrees was associated with increased mortality in 134 men with cirrhosis independent of MELD score [29]	Portable bedside test Non-invasive No radiation Inexpensive Rapid	Questionable accuracy as results affected by fluid retention, body mass index, and activity level [30]	High	Validation required in broader populations of cirrhotics Need to ascertain the degree of confounding by ascites
Ultrasound Quadriceps muscle diameter (cm) Muscle psoas index (psoas muscle diameter to height ratio)	Not defined	Psoas to height ratio was significantly associated with mortality in a cohort of 75 patients with decompensated cirrhosis (HR 0.825 (95% CI 0.701–0.973)) [31]	Rapid Inexpensive Non-invasive No radiation	Potential inter-operator variability Need further data on reproducibility	High	Requires validation in larger cohorts
Skeletal muscle index (height-adjusted muscle area) at the 3rd lumbar vertebrae using Computerized Tomography (CT) scan (CT L3 SMI—cm^2^/m^2^)	Men: <52.4 cm^2^/m^2^ Women: <38.5 cm^2^/m^2^ [2] Men: <50 cm^2^/m^2^ Women: <39 cm^2^/m^2^ [32]	Presence of sarcopenia was associated with mortality on multivariable analysis in 112 patients assessed for liver transplant (HR 2.21, *p* = 0.008) [2] A continuous inverse association was observed between SMI and mortality in 396 patients assessed for liver transplant (HR 0.95, *p* < 0.001) [32]	Able to accurately define muscle, visceral and subcutaneous fat Can assess muscle quality as well as quantity Not influenced by ascites Well-validated as a predictor of mortality	Expensive (scan and software) Radiation exposure May not be widely available (requires software) No clearly defined protocol for reproducibility No data on serial measurements	Low-moderate	Protocol to ensure standardization between patients and centers Data on serial measurements Validation of suggested cut-offs in broader cirrhotic populations
CT psoas muscle thickness adjusted for height (TPMT/height—mm/m) Psoas muscle area (PMA—mm^2^)	TPMT/height Men: <17.3 mm/m Women: <10.5 mm/m [33] PMA: Men: <1561 mm^2^ Women: <1464 mm^2^ [34]	Mortality risk was increased by 15% for each 1 unit decrease in TPMT/height in 376 patients assessed for liver transplant [3] PMA in the sarcopenic range was associated with reduced 12-month survival (59% vs. 94%, *p* < 0.001) in 256 patients with cirrhosis [34] Height-adjusted PMA was associated with mortality in women (HR 0.58, *p* = 0.002) but not men (HR 0.85, *p* = 0.09) in a cohort of 353 cirrhotics assessed for liver transplant [35]	Psoas muscle easily identifiable on CT scan No need for specific software to analyse scan Not influenced by ascites	Cost of scan Radiation exposure Less mortality data as compared to L3 SMI	Low-moderate	Further studies to establish correlation with mortality Data on serial measurements
Dual energy X-ray absorptiometry (DEXA) appendicular lean mass—height adjusted (APLM kg/m^2^)	Men: <7.26 kg/m^2^ Women: <5.45 kg/m^2^ [36] Men:<6.57 kg/m^2^ Women: <4.61 kg/m^2^ [37]	Muscle mass as measured by APLM was inversely correlated with mortality on multivariable analysis in a cohort of 144 men with cirrhosis (HR 0.44, *p* = 0.029) [38]	Highly precise and reproducible (coefficient of variation 0.5%) [39] Minimal radiation exposure Inexpensive Validated serial measures in clinical trial in cirrhosis [40]	Access issues Oedema may falsely elevated lower limb lean mass measure No data on mortality association in female cirrhotics	Moderate	Larger cohort studies required to assess impact of APLM on mortality, particularly in women
DEXA upper limb lean mass—height-adjusted (kg/m^2^)	Men: <1.6 kg/m^2^ [41]	Inverse association found between upper limb lean muscle and mortality in 420 men with cirrhosis (HR 0.27; 95% C.I 0.11–0.66; *p* = 0.004) [41]	Highly reproducible Low radiation exposure Inexpensive Less likely to be affected by oedema	Access issues No studies in female cirrhotics Evidence from single study of cirrhotics only	Moderate	Needs external validation in other cohorts of cirrhosis, particularly in women
Magnetic resonance imaging (MRI) Fat-free muscle area (FFMA: mm^2^) at the level of the superior mesenteric artery [42]	Men: FFMA <3197 mm^2^ Women: FFMA <2895 mm^2^	FFMA was an independent predictor of survival in 116 patients with cirrhosis undergoing TIPS stent placement (HR 0.92, *p* = 0.001)	No radiation exposure Can provide information on muscle quality (fat infiltration) as well as quantity	Expensive Access issues No clear protocols Few studies examining use in liver disease	Low-moderate	Needs further validation in patients with cirrhosis

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
