# Peer review of "Controversies in Diagnosing Sarcopenia in Cirrhosis—Moving from Research to Clinical Practice"

_nutrients, 2019, doi:10.3390/nu11102454_

Round 1
Reviewer 1 Report
Sinclair M, has performed an excellent review article on controversies in diagnosing sarcopenia in cirrhosis, moving from research to clinical practice. The author, after defining sarcopenia and its prevalence in cirrhotic patients, discusses what causes sarcopenia in cirrhosis and its relevance in these patients. Then, she discusses whether treating sarcopenia improve outcomes and commented on that there are little data on the accuracy of sarcopenia diagnosis in cirrhotic patients. She also questions whether examining sarcopenia as a binary phenomenon is the most appropriate way to analyze muscle mass or strength. Then, she reviewed the currently used methods to diagnose sarcopenia and where to next in sarcopenia research. She concludes that there is not a clear gold standard for diagnosing sarcopenia and the clinical and research consequences of the lack of standardized protocols for the different existing sarcopenia diagnostic tests. She finalizes demanding better ways to measure muscle mass and function to improve patient care.
Minor comments:
1. Introduction, lines 27-28:
The author defined sarcopenia in elderly patients as the loss of muscle mass and function (reference 1). This definition has recently been updated as low levels of measures for three parameters: muscle strength, muscle quantity/quality and physical performance as an indicator of severity (Cruz-Jentoft, et al. Age and Ageing 2019; 48:16-31). Please, update these comments and substitute reference 1 for the given reference.
Why is sarcopenia relevant in cirrhosis?, line 117:Please, substitute “WORSEN” for “worsen”
What methods are currently used to diagnose sarcopenia?The author should discuss two additional methods, Ultrasonography (US) and Magnetic Resonance Imaging (MRI), for diagnosing sarcopenia in any population including liver disease. I suggest you to include and discuss in the manuscript, among others, the following articles:
Hernández-Socorro CR, Saavedra P, López-Fernández JC, Ruiz-Santana S. Assessment of muscle wasting in long-stay ICU patients using a new ultrasound protocol. Nutrients. 2018 Dec 1;10(12). pii: E1849. doi: 10.3390/nu10121849.
Hari A, Berzigotti A, Štabuc B, Caglevič N. Muscle psoas indices measured by ultrasound in cirrhosis - Preliminary evaluation of sarcopenia assessment and prediction of liver decompensation and mortality. Dig Liver Dis. 2019 Sep 20. pii: S1590-8658(19)30780-7. doi: 10.1016/j.dld.2019.08.017.
Praktiknjo M, Book M, Luetkens J, Pohlmann A, Meyer C, Thomas D, Jansen C, et al. Fat-free muscle mass in magnetic resonance imaging predicts acute-on-chronic liver failure and survival in decompensated cirrhosis. Hepatology. 2018 Mar;67(3):1014-1026. doi: 10.1002/hep.29602. Epub 2018 Jan 24.
References:
From reference number 7 until reference 60 are displaced downwards because the author forgot to number the following one:
Sinclair M, Gow PJ, Grossmann M, Angus PW. Review article: sarcopenia in 302 cirrhosis--aetiology, implications and potential therapeutic interventions. Aliment Pharmacol 303 Ther 2016;43:765-777.
This reference follows reference number 6. Also, reference 16 in mixed-up with reference 17.
The following references are incompletely cited: number 45 (Liver Transpl 2019;25:1480-1487); number 48 (J Hepatol 2018;68:707-714); number 49 (Clin Mol Hepatol 2018;24:319-330).
Please, double-check all the manuscript references and modified them accordingly.
Author Response
Many thanks for your comments and suggestions. A point by point response is attached

Reviewer 2 Report
Dear Author
I read with great interest your article titled “Controversies in diagnosing sarcopenia in cirrhosis – moving from research to clinical practice”.
This is a review article exploring the hot topic of sarcopenia in cirrhosis.
The introduction about sarcopenia in cirrhosis and potential treatment benefits is well written and fluent, with relatively recent references.
After that, the structure of the review is a bit confusing.
The main topic, according to the title, is “Controversies in diagnosing sarcopenia in cirrhosis”. In the present form paragraphs 2, 3 and 4 have the importance of sarcopenia in cirrhosis as main topic. The following paragraphs 5,6 and 7 consider the diagnostic approaches.
However several techiniques are already mentioned in the introduction, and reported as evidences in section 3 “Why is sarcopenia relevant in cirrhosis? “ but not extensively explained until section 7.
Moreover I would change the title of the section according to the content, indeed the aim of the review is showing how a correct assessment of sarcopenia is important and not only why sarcopenia is relevent in cirrhosis, ie Why assessing sarcopenia is so important in cirrhosis?
Section 5 starts abruptly with CT scan, probably a small introduction about what the guidelines recommend for the diagnosis could be useful (this could alternatively be discussed in what is now section 7), and only after that exploring the evidences in liver disease (paragraph 3).
Moreover a way to present the different techniques could be starting with the less invasive and clinically used, i.e. handgrip strenght, to move to the more experimental ones.
These are mentioned partially in section 7 and extensively in Table 1. I think that this should be one of the first paragraph on the review (after section 2).
There is no mention to ultrasound please add some relevant and recent papers on this topic.
A potential scheme could be:
Introduction What causes sarcopenia in cirrhosis? What methods are currently used to diagnose sarcopenia? Why is sarcopenia relevant in cirrhosis? What do we want in a diagnostic test for sarcopenia? Is sarcopenia a binary phenomenon? Does treating sarcopenia improve outcome? Where to next in sarcopenia research? Conclusion
Minor comments
Line 117: is there a particular reason why “WORSEN” is in capital letters?
Please better clarify what is the MELD-sarcopenia score
Final considerations:
Overall the paper needs some improvements, but I appreciate the hard work.
Author Response
Many thanks for your comments and suggestions.
The main topic, according to the title, is “Controversies in diagnosing sarcopenia in cirrhosis”. In the present form paragraphs 2, 3 and 4 have the importance of sarcopenia in cirrhosis as main topic. The following paragraphs 5,6 and 7 consider the diagnostic approaches.
However several techniques are already mentioned in the introduction, and reported as evidences in section 3 “Why is sarcopenia relevant in cirrhosis? “ but not extensively explained until section 7.
Moreover I would change the title of the section according to the content, indeed the aim of the review is showing how a correct assessment of sarcopenia is important and not only why sarcopenia is relevent in cirrhosis, ie Why assessing sarcopenia is so important in cirrhosis?
Section 5 starts abruptly with CT scan, probably a small introduction about what the guidelines recommend for the diagnosis could be useful (this could alternatively be discussed in what is now section 7), and only after that exploring the evidences in liver disease (paragraph 3).
Many thanks for this sensible recommendation it is a sensible idea to re-structure the article and this has been done accordingly
Moreover a way to present the different techniques could be starting with the less invasive and clinically used, i.e. handgrip strenght, to move to the more experimental ones.
These are mentioned partially in section 7 and extensively in Table 1. I think that this should be one of the first paragraph on the review (after section 2).
Sensible comment. Altered as requested with further detail added as to currently used diagnostic techniques
There is no mention to ultrasound please add some relevant and recent papers on this topic.
Added as per Reviewer 1
A potential scheme could be:
Introduction What causes sarcopenia in cirrhosis? What methods are currently used to diagnose sarcopenia? Why is sarcopenia relevant in cirrhosis? What do we want in a diagnostic test for sarcopenia? Is sarcopenia a binary phenomenon? Does treating sarcopenia improve outcome? Where to next in sarcopenia research? Conclusion
Very good comments, I have altered the structure of the review article accordingly, and additional added further explanation to “current methods to diagnose sarcopenia”
Minor comments
Line 117: is there a particular reason why “WORSEN” is in capital letters?
This has been altered
Please better clarify what is the MELD-sarcopenia score
This has been further clarified
Final considerations:
Overall the paper needs some improvements, but I appreciate the hard work.

Round 2
Reviewer 2 Report
Dear Author
the manuscript has been considerably improved, the reading is now fluent and understandable also for people not expert in this field.